# Distributed non-parametric deep and wide networks

## Abstract

In recent work, it was shown that combining multi-kernel based support vector machines (SVMs) can lead to near state-of-the-art performance on an action recognition dataset (HMDB-51 dataset). In the present work, we show that combining distributed Gaussian Processes with multi-stream deep convolutional neural networks (CNN) alleviate the need to augment a neural network with handcrafted features. In contrast to prior work, we treat each deep neural convolutional network as an expert wherein the individual predictions (and their respective uncertainties) are combined into a Product of Experts (PoE) framework.

## Introduction

Recognizing actions in a video stream requires the aggregation of temporal as well as spatial features (as in object classification). These video streams, unlike still images, have short and long temporal correlations, attributes that single frame convolutional neural networks fail to discover. Therefore, the first hurdle to reach human-level performance is designing feature extractors that can learn this latent temporal structure. Nonetheless, there has been much progress in devising novel neural network architecture since the work of Karpathy et al. (2014). Another problem is the large compute, storage and memory requirement for analysing moderately sized video snippets. One requires a relatively larger computing resource to train ultra deep neural networks that can learn the subtleties in temporal correlations, given varying lighting, camera angles, pose, etc. It is also difficult to utilise standard image augmentation (like random rotations, shears, flips, etc.) techniques on a video stream. Additionally, features for a video stream (unlike static images) evolve with a dynamics across several orders of time-scales.

Nonetheless, the action recognition problem has reached sufficient maturity using the two-stream deep convolutional neural networks (CNN) framework (Simonyan & Zisserman, 2014). Such a framework utilises a deep convolutional neural network (CNN) to extract static RGB (Red-Green-Blue) features as well as motion cues by deconstructing the optic-flow of a given video clip. Notably, there has been plenty of work in utilising a variety of network architectures for factorising the RGB and optical-flow based features. For example, an inception network (Szegedy et al., 2016) uses $1 \times 1$ convolutions in its inception block to estimate cross-channel corrections, which is then followed by the estimation of cross-spatial and cross-channel correlations. A residual network (ResNet), on the other hand, learns residuals on the inputs instead of learning unreferenced functions (He et al., 2016). While such frameworks have proven useful for many action recognition datasets (UCF101, UCF50, etc.), they are yet to show promise where videos have varying signal-to-noise ratio, viewing angles, etc.

We improve upon existing technology by combining Inception networks and ResNets using a Gaussian Process classifier that is further combined in a product-of-expert (PoE) framework to yield, to the best of our knowledge, a state-of-the-art performance on the HMDB51 data-set (Kuehne et al., 2013). Under a Bayesian setting, our framework provide not only mean predictions, but also the uncertainty associated with each prediction. Notably, our work forwards the following contributions:

- We introduce a framework that allow for independent multi-stream deep neural networks, enabling horizontal scalability
- Ability to classify video snippets that have heterogeneity regarding camera angle, video quality, pose, etc.

- Combine deep convolutional neural networks with non-parametric Bayesian models, wherein there is a possibility to train them using less amount of data

- Demonstrate the utility of model averaging that takes uncertainty around mean predictions into account

## METHODS

In this section, we describe the dataset, the network architectures and the nonparametric Bayesian setup that we utilise in our four-stream CNN pillar network for activity recognition. We refer the readers to the original network architectures in Wang et al. (2016) and Ma et al. (2017) for further technical details. Utilising classification methodologies like AdaBoost, gradient boosting, random forests, etc. provide us with accuracies in the range of 5-55% for this dataset, for either the RGB or the optic-flow based features.

### DATASET

The HMDB51 dataset (Kuehne et al., 2013) is an action classification dataset that comprises of 6,766 video clips which have been divided into 51 action classes. Although a much larger UCF-sports dataset exists with 101 action classes (Soomro et al., 2012), the HMDB51 has proven to be more challenging. This is because each video has been filmed using a variety of viewpoints, occlusions, camera motions, video quality, etc. anointing the challenges of video-based prediction problems. The second motivation behind using such a dataset lies in the fact that HMDB51 has storage and compute requirement that is fulfilled by a modern workstation with GPUs – alleviating deployment on expensive cloud-based compute resources.

All experiments were done on Intel Xeon E5-2687W 3 GHz 128 GB workstation with two 12GB nVIDIA TITAN Xp GPUs. As in the original evaluation scheme, we report accuracy as an average over the three training/testing splits.

### INCEPTION LAYERS FOR RGB AND FLOW EXTRACTION

We use the inception layer architecture described in Wang et al. (2016). Each video is divided into $N$ segments, and a short sub-segment is randomly selected from each segment so that a preliminary prediction can be produced from each snippet. This is later combined to form a video-level prediction. An Inception with Batch Normalisation network (Ioffe & Szegedy, 2015) is utilised for both the spatial and the optic-flow stream. The feature size of each inception network is fixed at 1024. For further details on network pre-training, construction, etc. please refer to Wang et al. (2016).

### RESIDUAL LAYERS FOR RGB AND FLOW EXTRACTION

We utilise the network architecture proposed in Ma et al. (2017) where the authors leverage recurrent networks and convolutions over temporally constructed feature matrices as shown in Fig. 1. In our instantiation, we truncate the network to yield 2048 features, which is different from Ma et al. (2017) where these features feed into an LSTM (Long Short Term Memory) network. The spatial stream network takes in RGB images as input with a ResNet-101 (He et al., 2016) as a feature extractor; this ResNet-101 spatial-stream ConvNet has been pre-trained on the ImageNet dataset. The temporal stream stacks ten optical flow images using the pre-training protocol suggested in Wang et al. (2016). The feature size of each ResNet network is fixed at 2048. For further details on network pre-training, construction, etc. please refer to Ma et al. (2017).

### NON-PARAMETRIC BAYESIAN CLASSIFICATION

Gaussian Processes (GP) emerged out of filtering theory (Wiener, 1949) in non-parametric Bayesian statistics via work done in geostatistics (Matheron, 1973). Put simply, GPs are collection of random variables that have a joint Gaussian distribution,

$$
\begin{aligned}
\text{Obervation:} \quad & y \,|\, f, \phi \sim \prod_{i=1}^{n} p\left(y_i \,|\, f_i, \phi\right) \\
\text{GP Prior:} \quad & \mathrm{f}\left(x\right)|\,\theta \sim \mathcal{GP}\left(m\left(x\right), k\left(x, \tilde{x}\,|\,\theta\right)\right) \\
\text{Hyperprior:} \quad & \theta, \phi \sim \mathrm{p}\left(\theta\right)\mathrm{p}\left(\phi\right)
\end{aligned}
\tag{1}
$$

where, $k\left(x, \tilde{x}\,|\,\theta\right)$ is the kernel function parameterized by $\theta$; $\phi$ is the parameter of the observation model; $\mathrm{f}\left(x\right)$ is the latent function evaluated at $x$ i.e., the features. $y$ denotes the class of the input features and $\{\phi, \theta\} \in \chi$ denote the set of hyper-parameters.

For multi-class problem with a non-Gaussian likelihood (softmax; $p\left(y_i \,|\, f\left(x_i\right)\right) = \frac{\exp\left(f_i^{y_i}\right)}{\sum\limits_{c=1}^{51} \exp\left(f_i^c\right)}$), the conditional posterior is approximated via the Laplace approximation (Williams & Barber, 1998) i.e., a second order Taylor expansion of $\log \mathrm{p}\left(f\,|\,y, \theta, \phi\right)$ around the mode $\hat{f}$ as,

$$
\begin{aligned}
\mathrm{p}\left(f\,|\,\mathcal{D}, \theta, \phi\right) \quad &\approx \quad q\left(f\,|\,\mathcal{D}, \theta, \phi\right) = \mathcal{N}\left(f\,\Big|\,\hat{f}, \Sigma\right) \\
\hat{f} \quad &= \quad \arg\max_{f} \mathrm{p}\left(f\,|\,\mathcal{D}, \theta, \phi\right) \\
\Sigma^{-1} \quad &= \quad -\nabla\nabla \log \mathrm{p}\left(f\,|\,\mathcal{D}, \theta, \phi\right)\big|_{f=\hat{f}} = K_{f,f}^{-1} + W \\
W_{ii} \quad &= \quad \nabla_{f_i}\nabla_{f_i} \log \mathrm{p}\left(y\,|\,f_i, \phi\right)\big|_{f_i=\hat{f}_i}
\end{aligned}
\tag{2}
$$

$\mathcal{D}$ is the (input,output) tuple. After the Laplace approximations, the approximate posterior distribution becomes,

$$
\begin{aligned}
\hat{f}\,\Big|\,D, \theta, \phi \quad &\sim \quad GP\left(m_p\left(\tilde{x}\right), k_p\left(\tilde{x}, \tilde{x}'\right)\right) \\
m_p\left(\tilde{x}\right) \quad &= \quad k\left(\tilde{x}, X\right)\nabla \log p\left(y\,|\,f\right)\big|_{f=\hat{f}} \\
k_p\left(\tilde{x}, \tilde{x}'\right) \quad &= \quad k\left(\tilde{x}, \tilde{x}'\right) - k\left(\tilde{x}, X\right)\left(K_{f,f} + W\right)^{-1}k\left(X, \tilde{x}'\right)
\end{aligned}
\tag{3}
$$

Finally, we can evaulate the approximate conditional predictive density of $\tilde{y}_i$,

$$
p\left(\tilde{y}_i\,|\,D, \theta, \phi\right) \approx \int p\left(\tilde{y}_i\,|\,\tilde{f}_i, \phi\right)q\left(\tilde{f}_i\,\Big|\,D, \theta, \phi\right)d\tilde{f}_i
\tag{4}
$$

PRODUCT OF EXPERTS

For each of the neural network, we subdivide the training set into $K = 7$ sub-sets so that $K$ different GPs could be trained, giving us 28 GPs for the 4 deep networks (2 Inception networks and 2 ResNets) that we have trained in the first part of our training. We assume that each of the 7 GPs are independent, such that the marginal likelihood in our product of expert (PoE) becomes,

$$
\begin{aligned}
p\left(y\,|\,X, \chi\right) \quad &\approx \quad \prod_{k=1}^{K} p_k\left(y^{(k)}\,\Big|\,X^{(k)}, \chi\right) \\
p_k\left(y^{(k)}\,\Big|\,X^{(k)}, \chi\right) \quad &= \quad -\frac{1}{2}y^{(k)}\left(K_\theta^{(k)} + \sigma_\phi^2\mathbf{I}\right)^{-1}y^{(k)} \\
&\quad - \frac{1}{2}\log\left|K_\theta^{(k)} + \sigma_\phi^2\mathbf{I}\right|
\end{aligned}
\tag{5}
$$

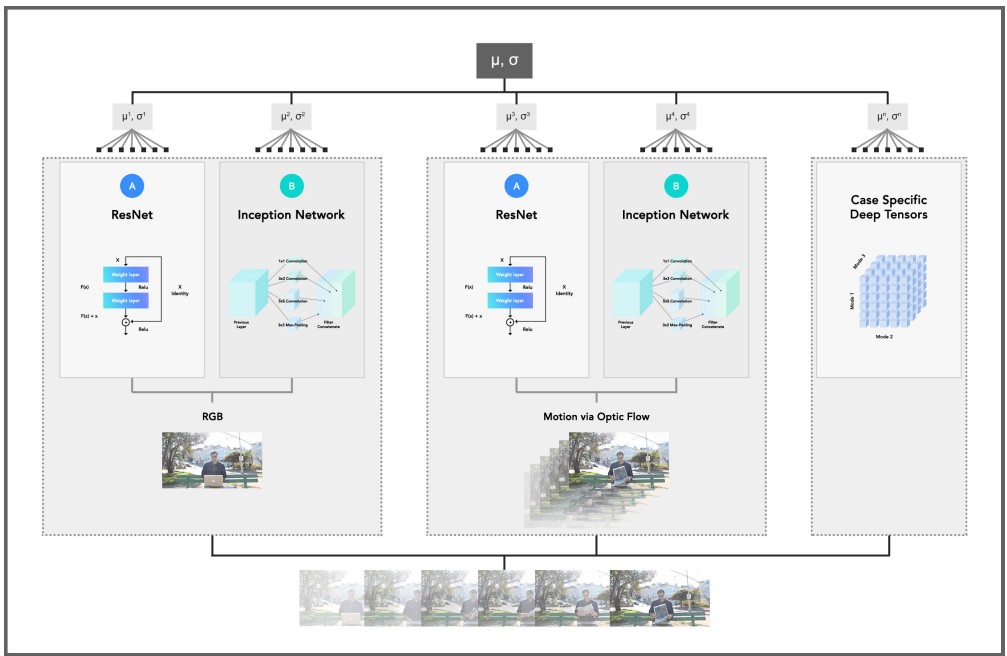

Figure 1: **The distributed non-parametric deep and wide network framework:** Each pillar represents either a single ultra-deep neural network or other feature tensors that can be learnt automatically from the input data. For action recognition, we factorize the static (RGB), and dynamic (optic flow) features using a ResNet and an Inception Network. Using the features of the last layer, we train seven Gaussian Processes for each of the network, which is combined under a Product-of-Experts (PoE) formalism. This hierarchy is then fused again to give us a prediction of action types.

What we have done is to reduce the computational expenditure from $\mathcal{O}(n^3)$ to $\mathcal{O}(n_k^3)$. Notice that unlike GPs with inducing inputs or variational parameters such a distributed GP does not require optimisation of additional parameters. Finally, a product-of-GP-experts is instantiated that predicts the function $f_*$ at the test point $x_*$ as,

$$
\begin{aligned}
p\left(f_* \,|\, x_*, D\right) &= \prod_{k=1}^{K} p_k\left(f_* \,\Big|\, x_*, D^{(k)}\right) \\
\mu_*^{poe} &= \frac{1}{\left(\sigma_*^{poe}\right)^2} \sum_k \sigma_k^{-2}\left(x_*\right) \mu_k\left(x_*\right) \\
\frac{1}{\left(\sigma_*^{poe}\right)^2} &= \sum_k \sigma_k^{-2}\left(x_*\right)
\end{aligned}
\tag{6}
$$

## RESULTS

We used 3570 videos from HMDB51 as the training data-set; this was further split into seven sub-sets, each with 510 videos. We select ten videos randomly chosen from each category, and each sub-set is non-overlapping. Based on seven sub-sets, seven GPs are trained on different features (RGB and Flow) from different Networks (TSN-Inception Wang et al. (2016) and ResNet-LSTM Ma et al. (2017)). In total, twenty-eight GPs are generated. The features for both the RGB and the optical flow were extracted from the last connected layer with 1024 dimension for the Inception network and 2028 for the ResNet network. The fusion is then performed both vertically (seven sub-sets) and horizontally (four networks). The accuracies of individual GPs and different fusion combinations (PoE) on split-1 are shown in Table 1. Fusion-1 represents the results from the fusion of seven GPs for each feature; Fusion-2 show the fusion result of RGB and Flow using different

Table 1: GP-PoE and SVM results for the HMDB51 data-set on split-1

| Accuracy [%] | Inception-RGB | Inception-Flow | ResNet-RGB | ResNet-Flow |
|---|---|---|---|---|
| GP-1 | 51.4 | 59.5 | 52.7 | 58.9 |
| GP-2 | 52.0 | 59.7 | 51.9 | 59.1 |
| GP-3 | 50.1 | 60.3 | 49.7 | 59.9 |
| GP-4 | 48.7 | 58.5 | 49.5 | 59.1 |
| GP-5 | 48.2 | 59.3 | 49.0 | 59.5 |
| GP-6 | 52.0 | 59.5 | 52.2 | 57.9 |
| GP-7 | 51.1 | 58.8 | 51.8 | 58.1 |
| Average | 50.5 | 59.4 | 51.0 | 58.9 |
| Fusion-1 | 54.6 | 62.6 | 54.8 | 61.6 |
| Fusion-2 | 69.7 | | 68.2 | |
| Fusion-all | **75.7** | | | |
| SVM-SingleKernel | 54.0 | 61.0 | 53.1 | 58.5 |
| SVM-MutliKernels-1 | 68.1 | | 63.3 | |
| SVM-MutliKernels-2 | 71.7 | | | |

Table 2: Accuracy scores for the HMDB51 data-set

| Methods | Accuracy [%] | Reference |
|---|---|---|
| Two-stream | 59.4 | Simonyan & Zisserman (2014) |
| Rank Pooling (ALL)+ HRP (CNN) | 65 | Fernando & Gould (2017) |
| Convolutional Two-stream | 65.4 | Feichtenhofer et al. (2016) |
| Pillar Networks + soft-max + cross-entropy | 67 | Sengupta & Qian (2017) |
| Temporal-Inception | 67.5 | Ma et al. (2017) |
| TS-LSTM | 69 | Ma et al. (2017) |
| ST-multiplier network | 68.9 | Feichtenhofer et al. (2017) |
| ST-ResNet + iDT | 70.3 | Ma et al. (2017) |
| Temporal Segment Network (2/3/7 modalities) | 68.5/69.4/71 | Wang et al. (2016) |
| ST-multiplier network + iDT | 72.2 | Feichtenhofer et al. (2017) |
| Pillar Networks + SVM-MKL | 72.8 | Sengupta & Qian (2017) |
| Pillar Networks + iDT + SVM-MKL | 73.0 | Sengupta & Qian (2017) |
| Pillar Networks + MIFS + SVM-MKL | 73.3 | Sengupta & Qian (2017) |
| Deep Convolutional Networks + GP-PoE | 73.6 | this paper |
| Deep Convolutional Networks + iDT + GP-PoE | 75.0 | this paper |

networks; Fusion-all shows the result by fusion of all the 28 GPs. The average result for three splits is displayed in Table 2. We also demonstrate the improvement when hand-crafted features like iDT are combined with dCNNs to yield an 1.4% improvement.

## DISCUSSION

Here, we make two contributions – (a) we build on recently proposed pillar networks (Sengupta & Qian, 2017) and combine deep convolutional neural networks with non-parametric Bayesian models, wherein they have the possibility of being trained with less amount of data and (b) demonstrate the utility of model averaging that takes uncertainty around mean predictions into account. Combining different methodologies allow us to supersede the current state-of-the-art in video classification especially, action recognition.

We utilised the HMDB-51 dataset instead of UCF101 as the former has proven to be difficult for deep networks due to the heterogeneity of image quality, camera angles, etc. As is well-known, videos contain extensive long-range temporal structure; using different networks (2 ResNets and 2 Inception networks) to capture the subtleties of this temporal structure is an absolute requirement. Since each network implements a different non-linear transformation, one can utilise them to learn very deep yet different features. Utilising the distributed-GP architecture then enables us to parcel-

late the feature tensors into computable chunks (by being distributed) of input for a Gaussian Process classifier. Such an architectural choice, therefore, enables us to scale horizontally by plugging in a variety of networks *as per requirement*. While we have used this architecture for video based classification, there is a wide range of problems where we can apply this methodology – from speech processing (with different networks) to natural-language-processing (NLP).

Ultra deep convolutional networks have been influential for a variety of problems, from image classification to natural language processing (NLP). Recently, there has been work on combining the Inception network with that of a Residual network such that the resulting network builds on the advantages offered by either network in isolation (Szegedy et al., 2017). In future, it would be useful to see how different are the features when they are extracted from Inception module, ResNet module or a combination of both. Not only this, a wide variety of hand-crafted features can also be augmented as inputs to the distributed GPs; our initial experiments using the iDT features show that this is indeed the case (a 1.4% improvement). Input data can also be augmented using RGB difference or optic flow warps, as had been done in Wang et al. (2016).

Also, the second stage of training, i.e., the GP classifiers work with far fewer examples than what a deep learning network requires. It would be useful to see how our framework performs on immensely large datasets such as the Youtube-8m data-set (Abu-El-Haija et al., 2016). Additionally, recently published Kinetics human action video dataset from DeepMind (Kay et al., 2017) is equally attractive, as pre-training, our framework on this dataset before fine-grained training on HMDB-51 will invariably increase the accuracy of the current network architecture.

The Bayesian product-of-GPs would suffer from a problem were we to increase the number of experts. This is because the precision of the experts adds up which leads to overconfident predictions, especially in the absence of data. In unpublished work, we have utilised generalised Product of Experts (gPoE) (Cao & Fleet, 2014) and Bayesian Committee Machine (BCM) (Tresp, 2000) to increase the fidelity of our predictions. These would be reported in a subsequent publication along with results from a robust Bayesian Committee Machine (rBCM) which includes the product-of-GPs and the BCM as special cases (Deisenroth & Ng, 2015).

For inference, we have limited our experiments to the Laplace approximation inference under a distributed GP framework. An alternative inference methodology for multi-class classification include (stochastic) expectation propagation (Riihimäki et al.; Villacampa-Calvo & Hernández-Lobato, 2017) or variational approximations (Hensman et al., 2015). Free-energy minimization is attractive simply due to lower computational overhead. Indeed, it comes with its problems such as underestimation of the variability of the posterior density, inability to describe multi-modal densities and the inaccuracy due to the presence of multiple equilibrium points. All being said, some of these problems are also shared by state-of-the-art MCMC samplers for dynamical systems. Due to the flexibility of utilising GPUs, both methods (variational inference and EP) can prove to be computationally efficient, especially for streaming data. Thus, there is a scope of future work where one can apply these inference methodologies and compare it with vanilla Laplace approximations, as utilised here.

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
