# OpenReview forum: "Distributed non-parametric deep and wide networks"
_ICLR.cc/2018/Conference — Reject_

### Official Review · AnonReviewer3 · 2017-11-27
**The paper proposes a classification system, that trains Gaussian Process based classifiers on a top of a number of neural networks, trained on two modalities (RGB and optical flow), and combines them  using a product of experts formulation. There is very low novelty and it is an applications paper which combines well known methods.**

**Rating:** 3
**Confidence:** 4

**Review:**

- The paper is fairly written and it is clear what is being done
- There is not much novelty in the paper; it combines known techniques and is a systems paper, so I
  would judge the contributions mainly in terms of the empirical results and messsage conveyed (see
  third point)
- The paper builds on a  previous paper (ICCV Workshops, https://arxiv.org/pdf/1707.06923.pdf),
  however, there is non-trivial overlap between the two papers, e.g. Fig. 1 seems to be almost the
  same figure from that paper, Sec 2.1 from the previous paper is largely copied
- The message from the empirical validation is also not novel, in the ICCVW paper it was shown that
  the combination of different modalities etc. using a multiple kernel learning framework improved
  results (73.3 on HMDB51), while in the current paper the same message comes across with another
  kind of (known) method for combining different classifiers and modality (without iDT their best
  results are 73.6 for CNN+GP-PoE)

---

### Official Review · AnonReviewer2 · 2017-11-28

**Rating:** 3
**Confidence:** 5

**Review:**

This paper, although titled "Distributed Non-Parametric Deep and Wide Networks", is mostly about fusion of existing models for action recognition on the HMDB51 dataset. The fusion is performed with Gaussian Processes, where each of the i=1,..,4 inspected models (TSN-Inception RGB, TSN-Inception Flow, ResNet-LSTM RGB, ResNet-LSTM Flow) returns a (\mu_i, \sigma_i), which are then combined in a product of experts formulation, optimized w.r.t. maximum likelihood.

At its current form this paper is unfit for submission. First, the novelty of the paper is not clear. It is stated that a framework is introduced for independent deep neural networks. However, this framework, the Gaussian Processes, already exists. Also, it is stated that the method can classify video snippets that have heterogeneity regarding camera angle, video quality, pose, etc. This is something characterizes also all other methods that report similar results on the same dataset. The third claim is that deep networks are combined with non-parameteric Bayesian models. That is a good claim, which is also shared between papers at http://bayesiandeeplearning.org/. The last claim is that model averaging taking into account uncertainty is shown to be useful. That is not true, the only result are the final accuracies per GP model, there is no experiment that directly reports any results regarding uncertainty and its contribution to the final accuracy.

Second, it is not clear that the proposed method is the one responsible for the reported improvements in the experiments. Currently, the training set is split into 7 sets, and each set is used to train 4 models, totalling 28 GP experts. It is unclear what new is learned by the 7 GP expert models for the 7 splits. Why is this better than training a single model on the whole dataset? Also, why is difference bigger between ResNet Fusion-1 and Resnet SVM-SingleKernel?

Third, the method reports results only on a single dataset, HMDB51, which is also rather small. Deriving conclusions from results on a single dataset is suboptimal. Other datasets that can be considered are (mini) Kinetics or Charades.

Forth,  the paper does not have the structure of a scientific publication. It rather looks like an unofficial technical report. There is no related work. The methodology section reads more like a tutorial of existing methods. And the discussion section is larger than any other section in the paper.

All in all, there might be some interesting ideas in the paper, specifically how to integrate GPs with deep nets. However, at the current stage the submission is not ready for publication.

---

### Official Review · AnonReviewer4 · 2017-12-04
**a specific architecture for action recognition, tested on one dataset -> limited contribution**

**Rating:** 3
**Confidence:** 4

**Review:**

Summary: the paper considers an architecture combining neural networks and Gaussian processes to classify actions in a video stream for *one* dataset. The neural network part employs inception networks and residual networks. Upon pretraining these networks on RGB and optical flow data, the features at the final layer are used as inputs to a GP classifier. To sidestep the intractability, a model using a product of independent GP experts is used, each expert using a small subset of data and the Laplace approximation for inference and learning.

As it stands, I think the contributions of this paper is limited:

* the paper considers a very specific architecture for a specific task (classifying actions in video streams) and a specific dataset (the HMDB-51 dataset). There is no new theoretical development.

* the elements of the neural network architecture are not new/novel and, as cited in the paper, they have been used for action classification in Wang et al (2016), Ma et al (2017) and Sengupta and Qian (2017). I could not tell if there is any novelty on this part of the paper and it seems that the only difference between this paper and Sengupta and Qian (2017) is that Sengupta and Qian used SVM with multi-kernel learning and this paper uses GPs.

* the paper considers a product of independent GP experts on the neural net features. It seems that combining predictions provided by the GPs helps. It is, however, not clear from the paper how the original dataset was divided into subsets.

* it seems that the paper was written in a rush and many extensions and comparisons are only discussed briefly and left as future work, for example: using the Bayesian committee machine or modern sparse GP approximation techniques, end-to-end training and training with fewer training points.

---

### Decision · Program_Chairs · 2018-01-29
**ICLR 2018 Conference Acceptance Decision**

**Decision:**

Reject

**Comment:**

Thank you for submitting you paper to ICLR. ICLR. The consensus from the reviewers is that this is not quite ready for publication.